

# Perilipin-related protein regulates lipid metabolism in *C. elegans*

Ahmed Ali Chughtai[1], Filip Kaššák[1], Markéta Kostrouchová[1,2],
Jan Philipp Novotný[1], Michael W. Krause[3], Vladimír Saudek[4],
Zdenek Kostrouch[1] and Marta Kostrouchová[1]

[1] Institute of Cellular Biology and Pathology, First Faculty of Medicine, Charles University in Prague, Albertov, Prague, Czech Republic
[2] Department of Pathology, Third Faculty of Medicine, Charles University in Prague, Ruská, Prague, Czech Republic
[3] Laboratory of Molecular Biology, National Institute of Diabetes and Digestive and Kidney Diseases, National Institutes of Health, Bethesda, MD, USA
[4] University of Cambridge Metabolic Research Laboratories, Wellcome Trust—Medical Research Council, Institute of Metabolic Science, Cambridge, UK

Corresponding authors
Vladimír Saudek, vs317@cam.ac.uk
Marta Kostrouchová,
marta.kostrouchova@lf1.cuni.cz

## ABSTRACT

Perilipins are lipid droplet surface proteins that contribute to fat metabolism by controlling the access of lipids to lipolytic enzymes. Perilipins have been identified in organisms as diverse as metazoa, fungi, and amoebas but strikingly not in nematodes. Here we identify the protein encoded by the *W01A8.1* gene in *Caenorhabditis elegans* as the closest homologue and likely orthologue of metazoan perilipin. We demonstrate that nematode *W01A8.1* is a cytoplasmic protein residing on lipid droplets similarly as human perilipins 1 and 2. Downregulation or elimination of *W01A8.1* affects the appearance of lipid droplets resulting in the formation of large lipid droplets localized around the dividing nucleus during the early zygotic divisions. Visualization of lipid containing structures by CARS microscopy *in vivo* showed that lipid-containing structures become gradually enlarged during oogenesis and relocate during the first zygotic division around the dividing nucleus. In mutant embryos, the lipid containing structures show defective intracellular distribution in subsequent embryonic divisions and become gradually smaller during further development. In contrast to embryos, lipid-containing structures in enterocytes and in epidermal cells of adult animals are smaller in mutants than in wild type animals. Our results demonstrate the existence of a perilipin-related regulation of fat metabolism in nematodes and provide new possibilities for functional studies of lipid metabolism.

## INTRODUCTION

Perilipins are regulatory proteins targeted to the surface of fat storage organelles called lipid dropplets (LDs) where they contribute to the regulation of lipid metabolism (*Brasaemle, 2007*). Functional perilipins (PLIN proteins encoded by the *PLIN* genes) (*Lu et al., 2001*) have been identified in very diverse organisms such as *Drosophila* (*Teixeira et al., 2003*), *Dictyostelium* (*Du et al., 2013*) and fungi (*Wang & St Leger, 2007*) and protein databases

list clear orthologues in diverse, non-plant eukaryota, including the simplest metazoan *Trichoplax adherens*, sponges, crustaceans, and choanoflagelates (UniProt proteins B3RRM2, I1GA14, G5DCP6, F2UJD9, respectively). In humans and other mammals, the PLIN family consists of five members (*Kimmel et al., 2010*) (Perilipin 1–5) with diverse tissue distribution, specificity, and partially redundant functions. Strikingly, no perilipin othologue has been identified in *C. elegans*, suggesting that nematode-specific lipid regulatory pathways might exist in this phylum and perhaps in others as well.

This unusual evolutionary gap in the perilipins prompted us to re-examine the *C. elegans* genome for a gene related to mammalian perilipin. We identify *W01A8.1* as the likely *C. elegans* orthologue of mammalian perilipin genes. We show that W01A8.1 is the previously unrecognized *C. elegans* homologue of vertebrate perilipins that possesses all functional domains characteristic for perilipins and functions in lipid metabolism at the level of lipid droplets.

The protein encoded by W01A8.1 in *C. elegans* is identified as Mediator Complex subunit 28 (MDT-28) in many protein databases (e.g., Pfam, UniProt, PIR, WormPep) (accessed on March 14, 2015), but the bioinformatics analysis reveals that this is a misannotation. We observe that protein isoforms expressed from *W01A8.1* are cytoplasmic proteins, residing predominantly on membranous structures of enterocytes and epidermal cells that have the characteristics of lipid droplets. We also show that transgene-encoded GFP fusion proteins of human Perilipins 1 and 2 localize in *C. elegans* similarly as W01A8.1::GFP on vesicular structures that are positive for lipid content. Furthermore, down regulation of *W01A8.1* by RNAi or its elimination lead to an altered appearance and behavior of lipid droplets prominently observed in the germline and in early embryos. Our results indicate that *C. elegans* can compensate for the loss of *W01A8.1* in all developmental stages except early embryos likely by additional fat degradation mechanisms.

Our data demonstrates that the perilipin-related regulation of fat metabolism is conserved in *C. elegans*, and provides a novel insight into early embryonic lipid management. This discovery offers promising possibilities for functional studies of lipid metabolism in a nematode model system.

## MATERIALS AND METHODS

### Sequence analysis

Perilipin orthologues and W01A8.1 sequences were extracted from UniProt, NCBI and OMA (omabrowser.org) databases. Chordate and nematode sequences were aligned separately using the T-Coffee algorithm (*Notredame, Higgins & Heringa, 2000*) (server tcoffee.crg.cat) and submitted to PSI-BLAST (*Altschul et al., 1997*) (E-value inclusion threshold $<10^{-3}$, 5 iterations) and HHpred (*Remmert et al., 2011*; *Biegert & Soding, 2008*) searches as implemented in MPItoolkit (toolkit.tuebingen.mpg.de). Repeat detection used HHrepID module in MPItoolkit. Alignments were displayed and analyzed in Jalview app (www.jalview.org).

## Strains, transgenic lines and genome editing

Wild type animals, N2 (var. Bristol), were used unless otherwise noted and all strains were maintained as described (*Brenner, 1974*). Transgenic lines were prepared using microinjections into gonads of young adult N2 hermaphrodites as described (*Tabara et al., 1999*; *Timmons, Court & Fire, 2001*; *Vohanka et al., 2010*). All injections also included mCherry co-injection markers: pCFJ90, pCJ104 and pGH8 (*Dickinson et al., 2013*).

To create mutants, we employed CRISPR/Cas9 system as described (*Dickinson et al., 2013*). The following plasmids were constructed: pCK001 targeting the sgRNA (+323) to the second exon of the *W01A8.1* gene (forward primer #7992), and pCK023 targeting the sgRNA (+1,372) to the sixth exon (forward primer #8078). The reverse primer was #7993. A scheme of known expressed isoforms listed in WormBase WS246 and the strategy for the disruption of *W01A8.1* gene is shown in Figs. S1 and S2. Primers used in this study are listed in Table S1.

The following transgenic lines regulated by *W01A8.1* natural promoter were prepared: *W01A8.1a/c::gfp* and *W01A8.1b::gfp* (containing the whole coding sequence of isoforms a and b). *W01A8.1* isoforms a and c have identical 3′ ends which both could be expressed from *W01A8.1a/c::gfp*. This construct also includes complete untagged isoform b. The GFP-tagged isoform a (plasmid pCK28 {$P_{W01A8.1}$::*W01A8.1(a)synth::gfp::unc-54 3′ UTR*}) was constructed by synthesizing the *W01A8.1a* sequence with modified codons to allow protection from CRISPR/Cas9 targeted sgRNA and prepared as a GeneArt® Strings™ DNA Fragment from Invitrogen (Invitrogen, Carlsbad, California, USA) and cloned using GeneArt® Seamless Cloning System (Invitrogen) into pPD95.75(NeoR). Schemes for isoforms expressed from *W01A8.1* gene and preparation of GFP tagged transgenes are given in Figs. S1 and S2.

Human *PLIN2* and *PLIN3* were cloned from a collection of anonymous unmarked samples (*PLIN2*), and from human peripheral lymphocytes (*PLIN3*) donated by a volunteer with a written consent in compliance with the legislation of the Czech Republic and European Union (Act No 372/2011 of 11. 11. 2011 on Health Care Services, Coll., Paragraph 81, section 1a and section 4a, which is in accordance with the declaration of Helsinki) and was approved by the Ethics Committee of the First Faculty of Medicine, Charles University in Prague (Ref. No. MZ13-UK1LF-KostrouchZdenek). Human PLIN1 optimized for *C. elegans* was prepared as a synthetic sequence requested as a GeneArt® Strings™ DNA Fragment from Invitrogen™.

Transgenic lines expressing human *PLIN1*, *PLIN2*, *PLIN3* tagged by GFP under *W01A8.1* natural promoter were prepared using N2 animals and animals with disrupted *W01A8.1*. Primers used for cloning *PLIN2* and *PLIN3* are listed in Table S1.

## Downregulation of gene expression by RNA interference

Downregulation of *W01A8.1* expression used the RNAi protocol of injection of dsRNA into gonads of young adult hermaphrodites as well as RNAi through feeding animals bacteria producing dsRNA as previously described (*Tabara et al., 1999*; *Timmons, Court & Fire, 2001*; *Vohanka et al., 2010*).

## Injection RNAi protocol

Double stranded RNA (dsRNA) was prepared for injection by *in vitro* transcription reactions (SP6/T7 Riboprobe® *in vitro* Transcription Systems; Promega, Madison, Wisconsin, USA) from opposing promoters and subsequent annealing of each single stranded RNA (ssRNA) product prior to injection. For RNAi directed against *W01A8.1*, BamHI or ApaI linearized pCK014 plasmid preparations were used in separate reactions to generate complementary ssRNA. After linearization, the DNA was phenol-chloroform extracted and ethanol precipitated. BamHI linearized DNA was transcribed using T7 RNA Polymerase while ApaI linearized DNA with SP6 RNA Polymerase. After *in vitro* transcription (∼2 h), equal volumes of sense and antisense RNA were incubated at 75 °C for 10 min and then cooled at room temperature for 30 min. Control RNAi was prepared from the promoter region of *nhr-60* as previously described (*Simeckova et al., 2007*). The dsRNA concentration was measured using a UV spectrophotometer and ∼1 μg/μl was used for injections.

## Feeding RNAi protocol

Nematode Growth Medium (NGM) agar plates were prepared according to standard protocols and were supplemented with Ampicillin (100 μg/ml final concentration) and isopropyl $\beta$-D-1-thiogalactopyranoside (IPTG) (1.5 mM final concentration). *E. coli* strain HT115 was transformed with pCK015 and control L4440 vector. After transformation, a single colony from each was used to inoculate LB medium with Ampicillin (100 μg/ml final concentration). The culture was grown to $OD_{600} \approx 1.0$; 900 μl of culture was poured onto NGM agar plates to completely cover the surface and 750 μl of the suspension was removed to leave 150 μl of the suspension on the plates. The bacteria were allowed to grow and were induced overnight at room temperate (∼22 °C).

## Fecundity and brood size assay

Fecundity measurement following RNAi (injection method) was conducted using a total of 50 young adult worms (25 control and 25 inhibited by RNAi specific for *W01A8.1*). Progeny was counted 24 h and again 48 h after injections. Brood size assay was performed for W01A8.1 disrupted animals and controls ($n = 15$ for each group). The progeny was determined during 6 days. The experiments were conducted at room temperature ∼22 °C.

   Fecundity measurement after RNAi using feeding protocol was performed over two generations to maximize the effect of knockdown. For this, a semi-synchronized population was isolated using standard WormBook (http://www.wormbook.org/) bleaching protocol. Hatched L1 stage worms were placed on NGM agar RNAi (*W01A8.1* specific and control) plates at ∼22 °C. Small, synchronized populations, of parents (P0) were transferred to fresh RNAi plates and allowed to lay progeny (F1). F1 generation animals were transferred to new RNAi plates and F2 generation was scored for a total of 21 F1 parents in each group. The experiment was repeated twice to confirm the results.

## RNA isolation and cDNA synthesis

Total RNA was extracted as previously described (*Vohanka et al., 2010*). Briefly, *C. elegans* washed, pelleted and re-suspended in re-suspension buffer with proteinase K. After lysis

TRIzol® Reagent (Invitrogen) was added to the mixture and the standard manufacturer's protocol was followed to obtain total RNA. Samples were then treated with DNaseI (New England Biolabs, Ipswich, Massachusetts, USA) and again TRIzol-chloroform extracted (Invitrogen) to obtain DNA free total RNA.

Human total RNA was also extracted using TRIzol® Reagent (Invitrogen) from peripheral blood lymphocytes and fat tissue. cDNA was prepared from total RNA by reverse transcription using standard protocols for the SuperScript® III First-Strand Synthesis System (Invitrogen) and oligo(dT) priming.

### Transcript quantification

Quantitative polymerase chain reaction (qPCR) was performed with cDNA prepared from total RNA isolated as described above, using the Roche Universal Probe Library technique (Hoffmann-La Roche, Basel, Switzerland). Primers and probes for determination of number of transcripts of *W01A8.1* are given in Table S1. Levels of *W01A8.1* expression were normalized against *ama-1*.

### Single worm PCR

Single animals were placed into 5 µl of worm lysis buffer (10 mM Tris–HCl pH 8.3, 50 mM KCl, 2.5 mM MgCl2, 0.45% NP-40, 0.45% Tween 20, 0.01% Gelatin and 500 µg/ml fresh proteinase K) in a PCR tube. Animals were frozen at −80 °C for 5 min before placing the tube into a thermal cycler and run under the following conditions: heat to 60 °C for 60 min followed by inactivation of proteinase K by heating to 95 °C for 20 min. Post-lysis, a PCR reaction mix (45 µl) targeting the template of choice was added and cycled for ∼35 times with Q5® Hot Start DNA polymerase (New England Biolabs).

### LipidTox staining

The lipid staining protocol was done as described (*O'Rourke et al., 2009*) with modifications. Approximately 200–500 animals were harvested from NGM plates with 1X PBS and washed several times to remove *E. coli* and pelleted at 1,500 × g. To the pellet, 500 µl 2X MRWB (160 mM KCl, 40 mM NaCl, 14 mM $Na_2$EGTA, 1 mM Spermidine 3HCl, 0.4 mM Spermine, 30 mM NaPIPES pH 7.4, 0.2% beta-ME) and 100 µl 20% paraformaldehyde were added and the volume was adjusted up to 1 ml with 1X PBS. Inverting the tube several times mixed the worms in solution after which it was allowed to fix for ∼60 min at room temperature with gentle shaking.

After fixation, animals were pelleted at 1,500 × g and washed 3 times with 1 ml Tris–HCl buffer (100 mM, pH 7.4). After the third wash, the supernatant was discarded down to 100 µl and 650 µl of Tris-HCL buffer was added followed by 250 µl of fresh/frozen reduction buffer (100 mM Tris-Cl pH 7.4, 40 mM DTT). Worms were then left shaking for ∼30 min at room temperature. After reduction, worms were washed 3 times in 1X PBS. After the final PBS wash, the volume was brought up to 500 µl and then 500 µl of LipidTox (Red) (1:500 dilution) (Invitrogen) was added to make a final volume of 1 ml. The final concentration of 1:1,000 dilution of LipidTox was used. The worms were left in the dark for at least 60 min with shaking before viewing.

## Microinjections

Microinjections of plasmids, DNA amplicons or dsRNA into gonads of young adult hermaphrodites were done using an Olympus IX70 microscope equipped with a Narishige microinjection system (Olympus, Tokyo, Japan).

## Microscopy

Fluorescence microscopy and Nomarski optics microscopy were done using an Olympus BX60 microscope equipped with DP30BW CD camera (Olympus, Tokyo, Japan).

## Coherent Anti-Stokes Raman Scattering microscopy (CARS)

For label-free CARS imaging the worms were anesthetized with levamisole and placed on a cover glass with a thin layer of 2% agarose. The CARS images were acquired with a Leica TCS SP8 CARS system (Leica Microsystems, Mannheim, Germany) consisting of a TCS SP8 confocal microscope combined with a picoEmerald laser (APE, Berlin, Germany) offering a fixed Stokes laser line of 1064.5 nm and a tuneable Pump line from an optical parametric oscillator (780 nm–940 nm). A HC PL IRAPO 40x water immersion objective was used for the imaging and CARS signal was detected with a non-descanned photon multiplier tube detector at the transmitted light side. For imaging of $CH_2$ vibration with Raman shift of 2,868 $cm^{-1}$ pump wavelength of 815.5 nm was used. The CARS signal was selected with a CARS2000 filter cube placed in front of the detector. Adult animals and embryos were completely scanned and recorded as stacks of focal planes. Recordings for quantitative analysis were done at fixed settings for mutant and wild type.

## Image analysis

Single focal planes (containing the highest number of CARS positive structures) from stacks of representative embryos and adult hermaphrodites were selected and analyzed using ImageJ computer program (http://imagej.nih.gov/ij/). Analysis for embryos was done on images of seven different embryos (seven mutant and seven wild type embryos) inside gravid hermaphrodites (only one and two cell early embryonic stages were chosen for comparison). Analysis of adult somatic tissue was performed on distal body region and we compared five different mutant adults with five different wild type adults. The images selected for analysis have been provided as supplement.

   Automatic particle counting feature of ImageJ program was used for determination of the number and area of CARS positive structures with manual thresholding as described on http://imagej.net/Particle_Analysis. Image area required for analysis was first selected (the area outside the selected zone was cleared) then the image was converted to an 8-bit scale. Manual threshold was applied with settings yielding the biggest number of individually recognizable structures (adult tissue threshold setting range was 36–200 and for embryos 11–13 to 200). Overlapping structures were separated using the "Watershed" command and also by manual line draw feature. "Analyze particles" command generated data sets containing the number and area of particles. Microsoft Excel 2003 was used to perform statistical analysis and two-tailed Student's $t$-test for determining the $p$-value.

Raw data sets are provided as supplement and labels in the Excel tables correspond to the marked images also provided as supplement.

## RESULTS

### Identification of a perilipin orthologue in *C. elegans*

We performed BLASTp searches with individual protein sequences of human perilipins that generated no significant hits to Nematoda sequences in the UniProt database, consistent with previous efforts that failed to identify a perilipin-related protein in this phylum. However, when a sequence alignment of chordate perilipins 2 and 3 (OMA database) was submitted as query in PSI-BLAST, the *C. elegans* protein W01A8.1a (Q23095_CAEEL) was identified as a highly significant hit ($E = 3 \times 10^{-13}$). A reciprocal PSI-BLAST search with the aligned closest nematode homologues of W01A8.1a identified chordate perilipins as strong hits with human Perilipin 2 (significance score $E = 10^{-53}$) appearing in the second iteration of the search. Similarly, HHpred profile-to-profile searches with human perilipin sequences as a query of the *C. elegans* proteome identified proteins coded by W01A8.1 (a, b or c) and reciprocally W01A8.1a showed profile homology to all human perilipins and the corresponding Pfam (*Punta et al., 2012*) perilipin profile (PF03036). Each available nematode proteome contained only a single such perilipin-related sequence, in stark contrast to the insect and chordate proteomes that had 2–5 perilipin paralogues. A sequence alignment of Plin2 and 3 from two selected vertebrates is compared with their nematode homologues (Fig. 1). Although the sequence-to sequence comparisons are not sufficient to unravel the sequence homology between vertebrate and nematode Plins, the similarity appear clearly in the profile-to sequence (PSI-BLAST) and profile-to-profile (HHpred) searches. We conclude that vertebrate Plins and nematode W01A8.1 are statistically highly significant homologues.

The alignment encompasses a substantial part of *C. elegans* and human sequences (e.g., 90% of W01A8.1 and 87% of Perilipin 2) and covers all three domains characteristic for perilipins (N-terminal PAT, imperfect amphiphilic 11-mer repeat (*Brasaemle, 2007*) and C-terminal four-helix bundle (*Hickenbottom et al., 2004*)) covering approximately amino acids 10–100, 125–190 and 220–380 respectively in W01A8.1a. As W01A8.1 and human perilipins appear to be the best mutual reciprocal PSI-BLAST and HHpred hits, W01A8.1 is a very good candidate for a *C. elegans* orthologue of perilipin.

Protein databases annotate W01A8.1 as Mediator Complex subunit 28, hence the official protein name assignment of MDT-28 in WormBase (WS246). Pfam database (*Punta et al., 2012*) based the Mediator 28 Hidden Markov model profile on a seed alignment of bovine and mosquito Mediator 28 sequences with W01A8.1. This very profile was probably used subsequently in all automatic annotations of the nematode sequences. However, no substantial homology between W01A8.1 and Mediator 28 exists as shown in the above searches. Since using the WormBase name of W01A8.1 (MDT-28) would be misleading, the gene is referred here by the cosmid name *W01A8.1*, which gives rise to at least three protein isoforms designated W01A8.1a, W01A8.1b, and W01A8.1c from at least seven different transcripts (*W01A8.1a.1, W01A8.1a.2, W01A8.1b.1, W01A8.1b.2, W01A8.1b.3,*

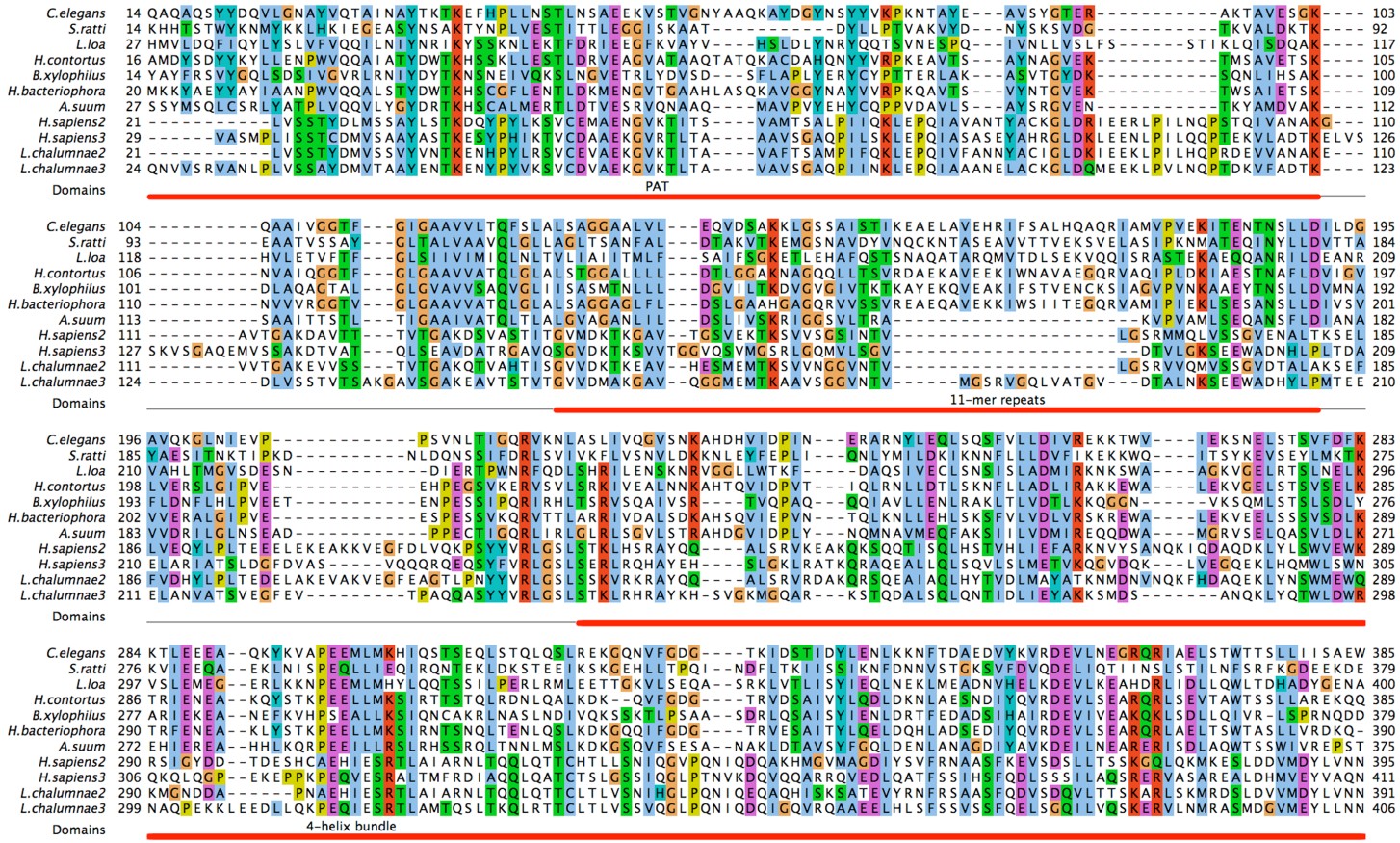

**Figure 1** **Identification of *C. elegans* protein W01A8.1a as a close homologue of vertebrate perilipin.** *C. elegans* protein W01A8.1a is compared with nematode homologues of pairwise sequence identity lower then 70% and with Plin2 and 3 from two diverse vertebrates. The three perilipin specific domains (indicated in red) were identified through homology with human Plin3. The six 11-mer repeats in W01A8.1a (positions 126–136, 137–147, 148–158, 159–169, 170–180 and 181–191) were established with HHrepID algorithm (*Biegert & Soding, 2008*). The N-terminal PAT domain is thought to interact with HSL. The central domain consisting of imperfect 11-mer repeats forming amphipathic helices is responsible for the main affinity to LDs and the C-terminal domain containing an apolipoprotein-like 4-helix bundle probably plays an additional role in the affinity to LDs and is known to interact with ABHD5 in mammalian Plin1 and 3 (*Brasaemle, 2007*). Alignment was done using T-coffee alignment of all available nematode sequences aligned with vertebrate Plin2 and 3 sequences in three iterations using ProfileAlign routine in MyHits suite (myhits.isb-sib.ch). Selected sequences from top to bottom: (Species, database identifier): *Caenorhabditis elegans*, Q23095; *Strongyloides ratti*, CACX01001972.1; *Loa loa*, E1G5Y0 and ADBU02007219.1; *Haemonchus contortus*, CDJ80228.1; *Bursaphelenchus xylophilus*, CADV01008520.1; *Heterorhabditis bacteriophora*, ES742365.1 and ACKM01001830.1; *Ascaris suum*, U1NU60; *Homo sapiens* 2, PLIN2_HUMAN; *Homo sapiens* 3, PLIN3_HUMAN; *Latimeria chalumnae* 2, H3AYC0; *Latimeria chalumnae* 3, GAAA01019375.1. Nucleotide sequences were translated with Wise2 program (*Birney, Clamp & Durbin, 2004*). Amino acid types are colored according to the Clustal scheme (jalview.org/help/html/colourSchemes/clustal.html).

*W01A8.1c.1, W01A8.1c.2*). The three protein isoforms are 415, 385, and 418 amino acid residues in length for isoform a, b, and c, respectively (Fig. S1). According to the *C. elegans* nomenclature, we suggest to rename *W01A8.1* as *Cel-plin-1* (isoform a, b, and c) and proteins Cel-PLIN-1 (isoform a, b, and c).

### *W01A8.1* protein products are cytoplasmic and reside primarily on lipid droplets

If the proteins encoded by *W01A8.1* act as perilipins, they would be expected to be associated with lipid droplets (*Kozusko et al., 2015*). To test this, we created translational reporter transgenes regulated by the putative endogenous promoter expressing isoform b and lines in which the genomic locus was tagged by an in-frame C-terminal GFP cassette. The second transgene, *W01A8.1a/c::gfp*, is likely to express not only high levels of a and c tagged isoforms, but also the native isoform b (Fig. S1). The translational fusion constructs resulted in high levels of cytoplasmic proteins present in intestinal and epidermal cells on vesicular structures with the characteristic appearance of lipid droplets. This pattern of expression and cellular distribution was observed beginning at the three-fold embryonic stage and continued throughout development to adulthood (Fig. 2). To confirm that the observed GFP-associated vesicular structures were indeed lipid droplets, transgenic animals were stained with the lipophilic reagent LipidTox as previously described (*O'Rourke et al., 2009*). The translational GFP fusion protein reporters were localized at the periphery of fat droplets that were LipidTox positive (Fig. 2).

### Human PLINs 1 and 2 label identical compartments as W01A8.1 proteins in *C. elegans*

We prepared transgenic *C. elegans* lines expressing human PLIN1, PLIN2 and PLIN3 fused to GFP and regulated by the *W01A8.1* promoter. PLIN1::GFP and PLIN2::GFP were localized on spherical cytoplasmic structures primarily in gut and epidermal cells (Figs. 3A, 3C, 3D and 3F) with identical appearance as W01A8.1 translational reporter GFP tagged proteins and *Drosophila* PLIN1::GFP expressed in *C. elegans* as reported by *Liu et al. (2014)*. PLIN3 expression was diffusely cytoplasmic and only faintly defined spherical structures (Figs. 3G and 3I). The structures clearly labeled with PLIN1::GFP and PLIN2::GFP were also positive in LipidTox staining (shown for PLIN2::GFP in Figs. 3J–3L). We conclude that W01A8.1 proteins are localized on the same structures as human PLIN1 and PLIN2.

### *W01A8.1* reduction-of-function alters the appearance of lipid droplets in early embryos and causes a reduction of brood size

To test the function of *W01A8.1*, we used RNAi done by germline injection and by feeding. *W01A8.1* RNAi made by microinjections and feeding resulted in a significantly smaller brood size, with approximately 30% less progeny. RNAi made by microinjections resulted in ∼52% reduced progeny laid in the first 24 h after microinjections and after 48 h ∼28% reduction in progeny laid compared to controls ($n = 260$, $n = 550$, for day one and $n = 1,000$, $n = 1,400$ for day two).

Repetition of knockdown by RNAi feeding over two generations confirmed this observation (Fig. S4). dsRNA feeding caused the *W01A8.1* specific group to produce ∼30% less larvae compared to controls, experiment was repeated twice independently with consistent findings. We confirmed, using RT-qPCR, that feeding based knockdown

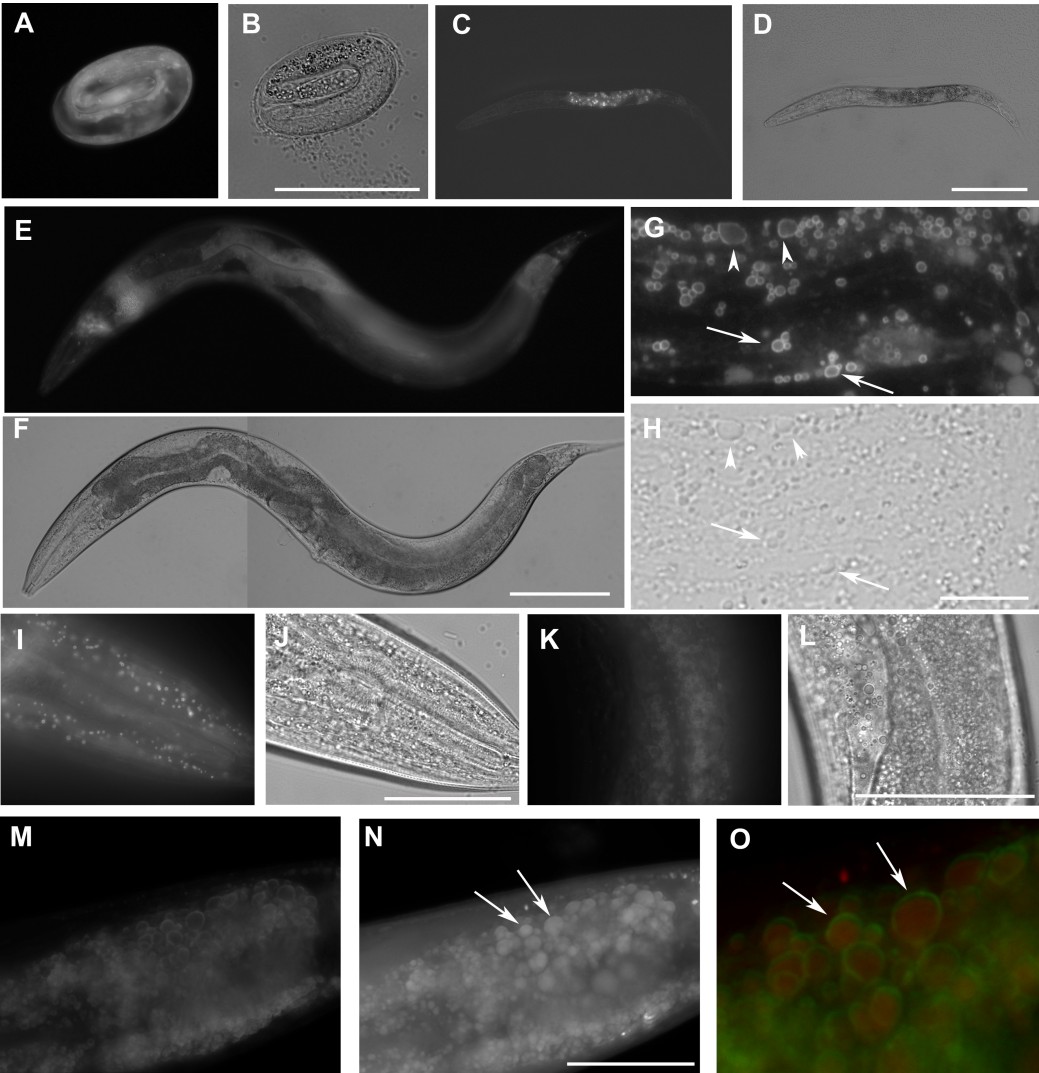

**Figure 2  The expression *W01A8.1::gfp* reporter genes in transgenic strains.** W01A8.1a/c::GFP is shown in (A, C, E, G), and (I), and corresponding areas in Nomarski optics are shown in (B, D, F, H) and (J). (A) The onset of expression of W01A8.1a/c::GFP in epidermal cells and in intestinal cells of three-fold embryo. (C) The expression of W01A8.1a/c::GFP in intestinal cells of an L2 larva. (E) and (G) W01A8.1a/c::GFP expression in epidermal cells and intestinal cells of a young adult hermaphrodite. (G) GFP fluorescence around lipid droplet-like structures in the intestine that are marked by arrows and arrowheads. Corresponding image in Nomarski optics is in (H). (I) A higher magnification the lipid droplet-like structures in epidermal cells labeled by W01A8.1a/c::GFP (shown in Nomarski optics in the J). (K) Lipid droplets of an unfixed intestine labeled by W01A8.1b::GFP (corresponding Nomarski image is in L). (M, N) and (O) Part of the intestine of an adult larva expressing W01A8.1b::GFP (M) with corresponding staining of lipid droplets by LipidTox (N). (O) LipidTox-positive lipid droplets (red) with W01A8.1b::GFP on the periphery (green) in this merged view. Bars represent 50 μm in (B, H, J, L) and (N) and 100 μm in (D) and (F).

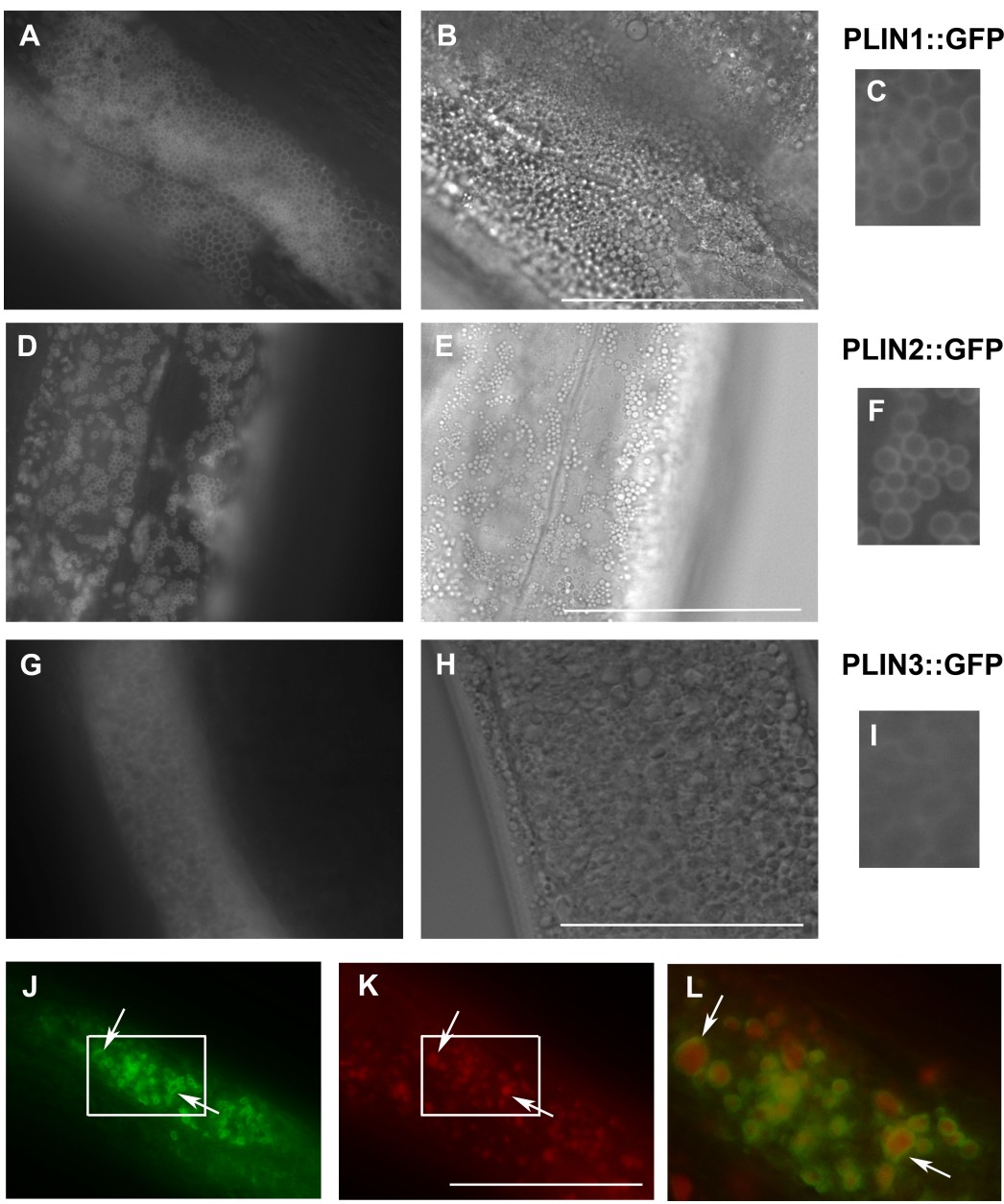

**Figure 3 Expression of human perilipins fused to GFP in *C. elegans*.** (A–C) Expression of human PLIN1::GFP in live transgenic *C. elegans*. PLIN1::GFP is localized on vesicles with an appearance of lipid droplets. PLIN2::GFP (D–F) is localized in transgenic animals on vesicular structures with an appearance of lipid droplets similarly as PLIN1::GFP. PLIN3::GFP (G, H and I) yields a more diffuse cytoplasmic pattern with faintly stained vesicular structures. (A, D) and (G) and details in (C, F) and (I) show GFP in fluorescence microscopy and (B, E) and (H) corresponding areas to (A, D) and (G) in Nomarski optics. (J, K) and (L) show PLIN2::GFP in fluorescence microscopy (J) in fixed *C. elegans* stained with LipidTox (K). The area indicated by the white rectangle in (J) and (K) is magnified and merged for co-localization of PLIN2::GFP (green) and LipidTox (red) in (L). Arrows indicate lipid droplets clearly marked by GFP with the LipidTox positive content. Bars represent 50 μm.

(represented in Fig. S4) resulted in approximately 45% decrease in *W01A8.1* transcripts (data not shown).

Staining of adult hermaphrodites with LipidTox (after formaldehyde fixation) revealed larger lipid droplets in early embryos derived from adults inhibited for *W01A8.1* (Figs. 4A and 4B) compared to controls (Figs. 4C and 4D).

## Targeted disruption of *W01A8.1* results in early embryonic defects but not lethality

In order to eliminate the W01A8.1 function completely, we designed a CRISPR/Cas9-mediated gene editing approach to eliminate almost the entire coding region (Fig. S2). We also included a rescuing plasmid consisting of isoform a that was prepared as cDNA synthesized *in vitro* using synonymous codons (*W01A8.1(a)synth::gfp*) that is protected against CRISPR/Cas9 targeted editing but allows the production of the wild type isoform a at the protein level. Lines that expressed the GFP fusion transgene were morphologically normal and W01A8.1(a)synth::GFP was found on lipid droplet-like structures as expected (Figs. 4E and 4G) that also stained positive by LipidTox (Figs. 4F and 4G). This transgenic strain yielded lines either carrying or losing the rescuing transgene in the background of a disrupted endogenous *W01A8.1*. The elimination of W01A8.1 was easily monitored by PCR (Fig. S3). Surprisingly, animals with the deleted *W01A8.1* locus that lost the extra-chromosomal rescuing array were able to reproduce normally. From several lines that had a confirmed disruption of *W01A8.1* and a confirmed loss of the extrachromosomal array, the line CK123 (KV001) was selected and used for subsequent analyses. As was observed in *W01A8.1* RNAi embryos, loss of *W01A8.1* activity resulted in the formation of large LipidTox-positive structures (Figs. 4H and 4I) that were clearly bigger than droplets observed in control embryos using the same protocol (Figs. 4C and 4D). These large lipid-containing structures were observable in live mutant embryos (Fig. 4J) but not in wild type embryos (Fig. 4K) using Nomarski optics. Viewing through multiple focal planes in live, developing embryos lacking *W01A8.1* showed that these large lipid droplets are present in embryos during the early mitotic divisions and were localized around the nucleus. Staining with LipidTox (after fixation) confirmed the lipid content in the vesicular structures arranged around dividing nucleus (Figs. 4L and 4M). These large vesicles persist through the two-cell stage, disappearing in most embryos with more than 6 cells. On fixed embryos stained with LipidTox, larger than wild type lipid droplets are visible until late embryonic stages, including three fold embryos.

In order to visualize lipid-containing structures in *W01A8.1* null mutants and in controls *in vivo*, we used CARS microscopy (done with kind help from Dr. Zhongxiang Jiang, Leica Microsystems, Mannheim, Germany). The CARS systems allow visualization of lipids of specific categories by tuning into symmetric $CH_2$ vibrations of specific fat composing molecules (*Zumbusch, Langbein & Borri, 2013*). CARS microscopy clearly confirmed the formation of large lipid containing vesicles in early embryos and allowed detailed analysis of the *W01A8.1* null phenotype. CARS microscopy also confirmed the gradual increase of the size of lipid containing structures during oogenesis (Figs. 5A

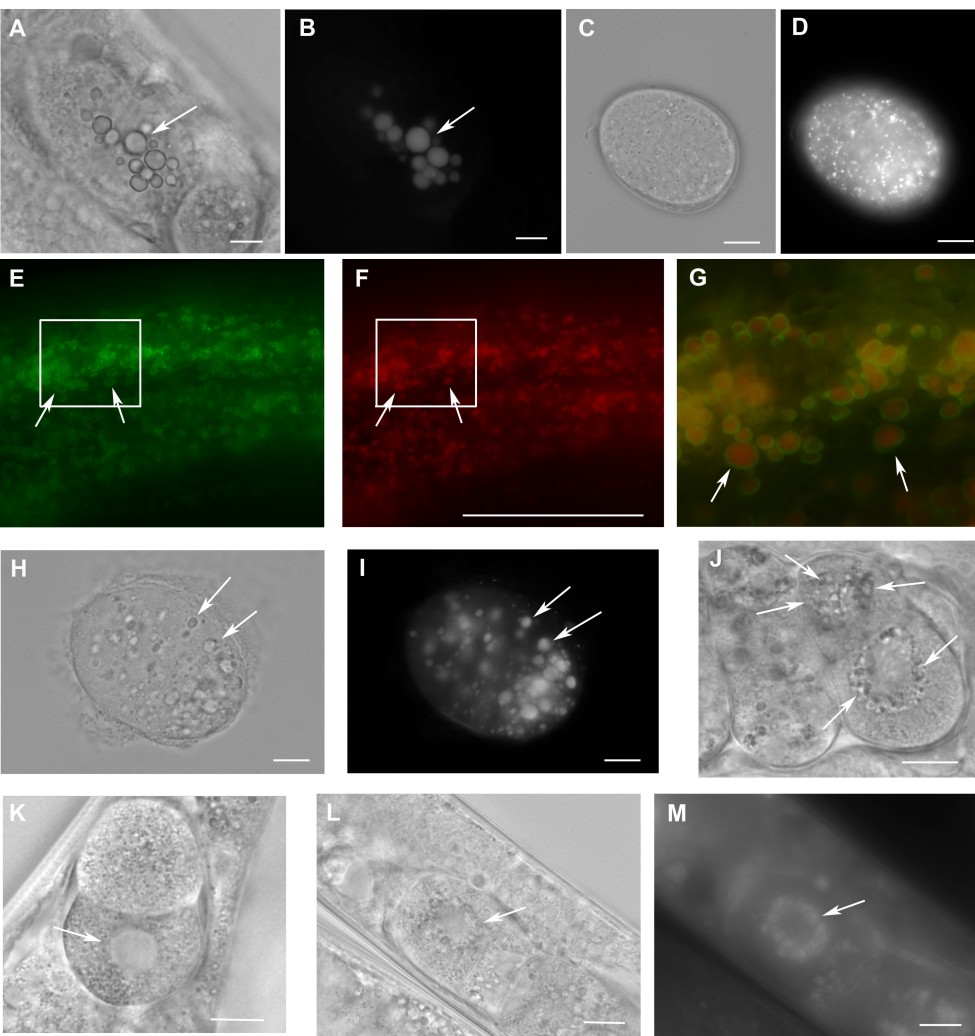

**Figure 4 Loss of *W01A8.1* function results in abnormal lipid droplet appearance.** (A) and (B) An embryo from a hermaphrodite inhibited for *W01A8.1* function by RNAi. Large lipid droplets stained by LipidTox (B) are visible also in Nomarski optics (A) in contrast with a control embryo that has only small and more evenly distributed lipid droplets (C—Nomarski optics and D—LipidTox staining). (E–J, L) and (M) Images of structures observed in animals with disrupted *W01A8.1*. (E) and (F) show structures with the appearance of lipid droplets in the intestine of an animal with disrupted W01A8.1 balanced with the synthetic transgene *W01A8.1(synth)::gfp*. GFP tagged synthetic W01A8.1a is localized on lipid droplets-like vesicular structures (E). (F) Shows the same area stained with LipidTox. (G) Shows in magnification a merged image of the area indicated by white rectangles in (E) and (F). Arrows indicate W01A8.1(synth)::GFP labeled lipid droplets (green) positive for lipids in LipidTox staining (red). (H) and (I) show an embryo of a parent with disrupted *W01A8.1* that had confirmed loss of the balancing transgene. Large LipidTox stained droplets are visible in Nomarski optics (H) as well as in LipidTox staining (I). Their enlargement is clearly visible in comparison with the wild type embryo shown in (C) and (D). (J) and (K) are images of live animals. (J) Shows an embryo with disrupted *W01A8.1* and confirmed loss of the balancing transgene. Large vesicular structures are formed around the dividing nucleus (arrows). (K) Shows a control embryo with normal appearance of the nuclear periphery (arrow). (L) and (M) show a one cell embryo from a parent with disrupted *W01A8.1* and confirmed loss of extrachromosomal array containing *W01A8.1(a)synth::gfp* after fixation and staining by LipidTox with large lipid droplets around the dividing nucleus visible in Nomarski optics (L) and positive for lipids in LipidTox staing (M) indicated by arrows. Bars represent 10 μm.

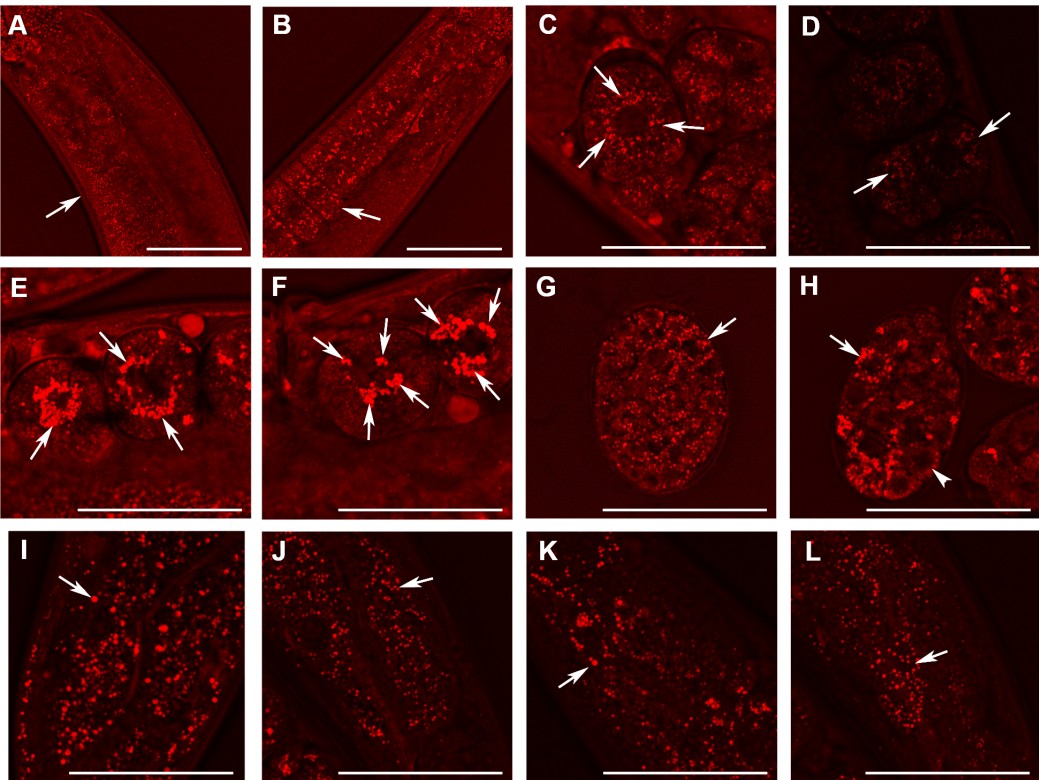

**Figure 5** **Analysis of lipid containing structures in live wild type animals and in animals with disrupted *W01A8.1* by CARS microscopy.** CARS microscopy was performed using constant magnification and intensity settings (20% laser intensity) with the exception of control embryos (C and G) that were examined at 30% laser intensity since the lipid content was lower in wild type. The brightness of the entire figure was digitally enhanced using Adobe Photoshop brightness setting (+150 units) for better visibility of structures. Arrows indicate lipid containing structures in paired panels. (A) The germline of a wild type adult hermaphrodite animal with small lipid containing structures in oocytes and an increase in their number and size during oogenesis. (B) The germline of a mutant hermaphrodite animal. Lipid containing structures are bigger compared to the control animal yet distributed evenly in mutant oocytes. (C) Lipid containing structures localize around the nucleus in one cell wild type embryos. This tendency of the association of the lipid containing structures with nuclear periphery can be seen also during later developmental stages in wild type embryos (D). (E, F) and (H) Enlarged lipid containing structures arranged around the nuclei in mutant embryos. (F) The formation of clusters of lipid containing structures on the periphery of nuclei. (G) Shows a wild type embryo at later stage of the development. (H) The lipid containing structures progressively diminish in size in mutant embryos during later stages of embryonic development (arrowhead). (I) and (J) Lipid containing structures in enterocytes of wild type (I) and mutant (J) animals. In contrast to embryos, which exhibit higher CARS signal and bigger lipid containing structures in mutant animals, gut cells in adult animals show the opposite, that is, a reduced fat-related CARS signal and smaller lipid containing structures in mutant animals. Similarly, lipid containing structures in epidermal cells (K and L) are bigger in wild type animals as shown in the (K) and smaller in mutant animals (L). Bars represent 50 μm.

and 5B), the sudden re-localization of these structures to the periphery of the dividing nucleus in the first embryonic division (Fig. 5E), and the propagation of this phenotype, although with gradually diminishing appearance, throughout embryonic development (Figs. 5F and 5H). CARS microscopy detected this phenomenon also in wild type animals, although the size of lipid containing structures was smaller making the phenomenon of

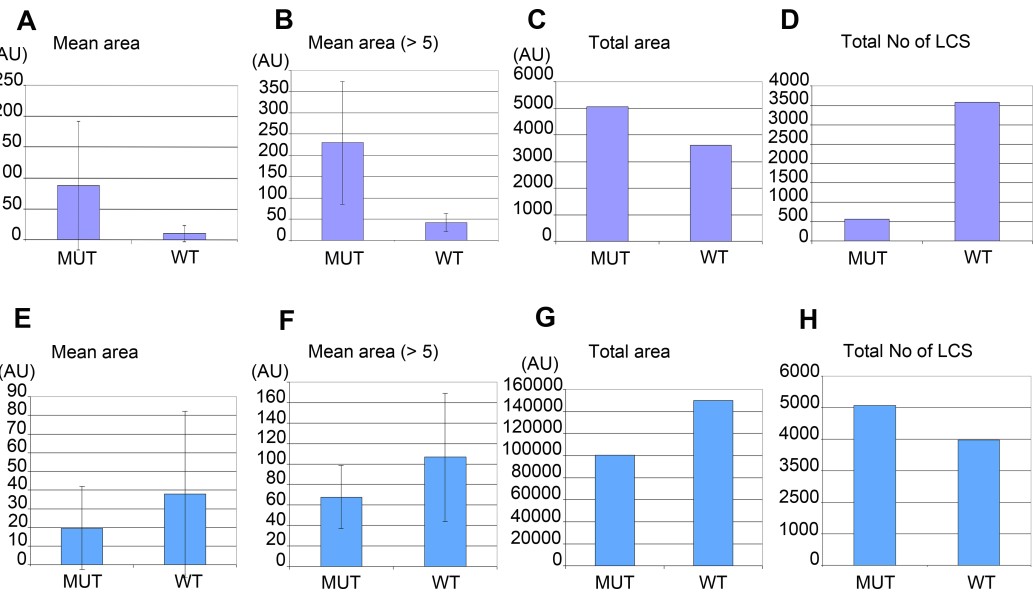

**Figure 6 Morphometric analysis of lipid containing structures by CARS microscopy in wild type and *W01A8.1* null animals.** Morphometric analysis was performed on CARS positive structures in single focal plane images acquired from representative mutant and wild type animal image stacks. (A–D) compare early embryos from mutant (MUT) and wild type (WT) animals using seven representative CARS images. (A) shows the mean area of all individually recognizable structures and (B) shows the mean area of all individually recognizable structures with an area bigger or equal to 5 arbitrary units (derived from pixels at the same settings). (C) compares the total area of all CARS positive structures in mutant and wild type embryos while (D) compares the total number of individually recognizable CARS positive structures (lipid containing structures—LCS) in the same embryos. (E–H) compare adult somatic tissue (tail region) from mutants and wild type hermaphrodites using five representative CARS images. (E) shows the mean area of all individually recognizable CARS positive structures and (F) shows the mean area of all individually recognizable structures with an area bigger or equal to 5 arbitrary units. (G) compares the total area of all CARS positive structures in mutant and wild type tail regions while (H) compares the total number of individually recognizable CARS positive structures (LCS) in the same regions. Vertical bars in (A, B, E) and (F) represent Standard Deviation. The results presented in (A, B, E) and (F) are statistically significant in two-tailed Student's $t$-test ($p < 0.0001$).

the sudden re-localization of lipid containing structures less obvious (Figs. 5C, 5D and 5G) than in *W01A8.1* null embryos. In contrast to embryos, lipid-containing structures in intestinal and epidermal cells of adult *W01A8.1* null mutants (Figs. 5J and 5L) were smaller than lipid-containing structures in control animals (Figs. 5I and 5K).

Morphometric analysis confirmed that *W01A8.1* null mutant embryos contained larger lipid positive structures recognized by CARS microscopy (Figs. 6A and 6B) that represent a larger total area (Fig. 6C), as determined by quantitating individual focal planes images. Morphometry revealed many small structures with area 1–4 AU (arbitrary units) with the provided threshold settings (Figs. 6A and 6E). There was a clear inverse relation in the number of large and small structures (with area <5 AU) for embryos as well as for adult tissues. The analysis in Figs. 6B and 6F shows that inclusion of small structures into analysis does not significantly affect the results but affects only standard deviation of the particle size distribution indicating that the results are independent on the setting of the limit for the size of lipid containing structures. The probability of the results were assayed using

Student's *t*-test and found to be statistically significant as the probability of this result, assuming the null hypothesis (no difference between control and experimental sets) was less than 0.0001.

Despite the fact that there were a larger number of individually recognizable lipid containing structures in wild type embryos (Fig. 6D), the mean area of these structures in mutants was considerably larger (Fig. 6C). In contrast, adult mutant animals contained smaller, more numerous lipid-containing structures (Figs. 6E, 6F and 6H) that covered a smaller total area (Fig. 6G) (and therefore volume) compared to wild type controls.

The morphometric analysis confirms that there is more CARS positive signal and therefore most likely more fat in *W01A8.1* null embryos (despite lower threshold used for analysis of wild type embryos) and less CARS positive signal (and less fat) in adult tissues of *W01A8.1* null animals compared to controls.

The analysis of the number of progeny laid by animals lacking W01A8.1 in comparison to wild type animals showed a decrease of progeny in mutant animals statistically significant on the day 3 (Fig. S5).

## DISCUSSION

Lipolysis is a tightly regulated cellular process in which triacylglycerol fatty acids (TAG) are degraded into free fatty acids (FFA) and glycerol (G) with intermediates of diacylglycerol (DAG) and monoacylglycerol (MAG). The function and regulation of three key lipases (adipose triglyceride lipase (ATGL), hormone-sensitive lipase (HSL) and monoglyceride lipase (MGL) have been studied in great detail in mammalian adipocytes (reviewed in *Lass et al., 2011*). Multidomain and multifunctional LD coating proteins, the perilipins, mediate the access of ATGL and HSL to LDs. Briefly (Fig. 7), the phosphorylated N-terminal domain PAT in perilipin interacts with HSL and brings it in contact with lipid droplets (LDs) (*Shen et al., 2009*). At the same time, the C-terminal phosphorylation (controlled by the kinase PKA) releases a specific activator of ATGL named ABHD5 without which ATGL remains inactive in the cytoplasm. The final step of the glycolysis is catalyzed by MGL. Conversely, unphosphorylated perilipin blocks lipolysis in the basal fed state by blocking the access of lipolytic enzymes to the fat stored in LDs. Both HSL and ATGL are the rate-limiting enzymes needed for fatty acids mobilization (*Schweiger et al., 2006*). A variation of this regulatory process, although less well understood in detail, exists in other cells and organisms. Most organisms so far studied contain several perilipin genes, complicating the analysis of complete perilipin loss-of-function.

Clear orthologues of ATGL, HSL, MGL, ABHD5 and catalytic and regulatory subunits of PKA have been identified in *C. elegans* (ATGL-1, HOSL-1, LID-1, KIN-1, KIN-2 respectively (*Lee et al., 2014*; *Xie & Roy, 2015*) (Fig. 7). The MGL orthologue remains to be identified but several un-annotated homologous proteins exist (*Birsoy, Festuccia & Laplante, 2013*). A recent careful and elegant study of ATGL function and regulation (*Lee et al., 2014*) revealed that the process in *C. elegans* was almost identical to that found in mammalian adipocytes. Even the degradation of ABHD5 in the proteasome

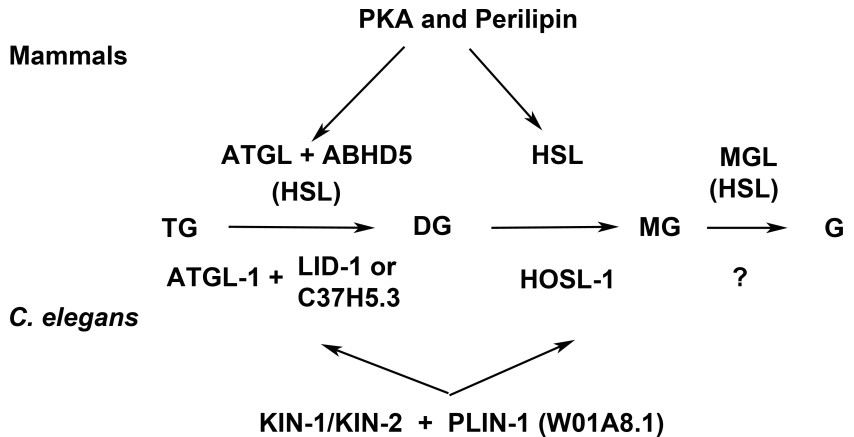

**Figure 7** **Enzymes and regulatory proteins involved in lipolysis (Adapted from *Lass et al., 2011*).** Mammalian proteins are indicated above the arrows and their *C. elegans* orthologues (*Lee et al., 2014*) below. Triacylglycerol (TAG) is progressively hydrolysed to diacylglycerol (DAG), monoacylglycerol (MAG) and glycerol (G) by lipases specific for each of these steps: adipose triacylglycerol lipase (ATGL), hormone-sensitive lipase (HSL) and finally monoacylglycerol lipase (MGL). LID-1 and C37H5.3 were proposed to be orthologues of ABHD5/CGI58 in *C. elegans* (*Lee et al., 2014*; *Xie & Roy, 2015*). HSL also shows some activity in the first and third step. The access of ATGL and HSL to lipid droplets is regulated by perilipin, which is under the control of protein kinase A (PKA). W01A8.1 is proposed to be a perilipin orthologue in the present work.

(*Dai et al., 2013*) is mirrored in *C. elegans* (*Lee et al., 2014*). The glaring difference in fat storage and metabolism seemed to be the absence of perilipin in nematode genomes.

Here we have established that *C. elegans* possesses a close homologue of perilipin that is intimately involved in the regulation of lipid metabolism. Although the sequence alignment of *C. elegans* and human homologues of perilipin does not appear visually very informative, the underlying evolutionary conserved homology is statistically very significant. Perilipin is a scaffolding protein allowing co-evolution of interacting domains and divergence of non-docking sequences, thus the function can be conserved even with limited amino acid conservation across species. This evolutionary plasticity was already apparent in the alignment of the human perilipin paralogues where only the knowledge of the three-dimensional structure enabled observations of the similarities in the C-terminal domains (*Hickenbottom et al., 2004*). The nematode sequences have diverged beyond the point where pairwise comparisons used in routine searches can reveal homology, hence the difficulty in identifying the nematode orthologues. Only rigorous statistical analysis of the hidden Markov profiles of a great number of diverse sequences made it possible to identify the conserved domain composition.

The nematode perilipin-related protein W01A8.1 contains all three major perilipin features: N-terminal PAT domain, amphipathic region composed of imperfect helical repeats and C-terminal apolipoprotein-like four-helix bundle. In mammals, the first two domains are known to be responsible for the interaction with HSL and LDs respectively and the ATGL interaction region resides in the C-terminus following the bundle. The function of the bundle is still unclear but its stability probably fine-tunes the solubility and the affinity to LDs (*Brasaemle, 2007*). All these functions will have to

be investigated in the isoforms of W01A8.1 in the future. The repeats are confirmed by analysis of internal homology using the HHrepID algorithm and the helical composition by secondary structure prediction. The bundle appears not to be stabilized by $\beta$-sheets as in human Perilipin 3 as revealed by the absence of the homology in the C-terminal region; the $\beta$-sheets are similarly absent in Perilipin 1.

Our findings are consistent with a proteomic study that found that W01A8.1b is among the most abundant proteins associated with LDs (*Zhang et al., 2012*). Similarly, perilipins are abundant proteins on mammalian LDs, although the distribution and proportion of the individual isoforms changes depending on the cell type and metabolic state (*Brasaemle et al., 2004*). Perilipins are widely used as general markers of LDs and it seems that W01A8.1a or b can be exploited for the same purposes; human PLIN1 was recently proposed as a marker for LDs in *C. elegans* (*Liu et al., 2014*).

Surprisingly, in laboratory conditions *C. elegans* can overcome the complete loss of the perilipin-related protein W01A8.1, presumably by activating perilipin-independent lipid degradation. Previous work has shown that an additional lipid degradation pathway, autophagy, was important for lipid metabolism in *C. elegans* (*Lapierre et al., 2013*), mammals (*Singh et al., 2009*) as well as in yeast (*Van Zutphen et al., 2014*). Similarly in *Drosophila*, which has two perilipins (*plin1* and *plin2*), the double mutants are viable but have small lipid droplets. This suggests that perilipins are required for growth or maintenance of lipid droplets, but are dispensable for lipolysis (*Beller et al., 2010*; *Bi et al., 2012*). The abnormal LD behavior, but viability, of *W01A8.1* null animals strongly suggests a regulatory role for the nematode perilipin-related protein in the regulation of fat metabolism that is similar to perilipins in other phyla. Taking in account the opposing phenotypes of *Drosophila plin1* and *plin2* loss of function regarding to lipid droplet size, it is intriguing to speculate that individual W01A8.1 splice forms may support distinct functions as well. It is also possible that some functions of W01A8.1 protein forms may be related to a proposed ancestral protein acting differently than mammalian perilipins as was suggested by *Beller et al. (2010)*. A possibility of the existence of an ancestor protein with wider, less specific functionality may be also considered if such a parallel to enzymatic activities of ancestral proteins (*Hujova et al., 2005*) is taken in account.

CARS microscopy allowed a detailed analysis of lipid containing structures in wild type and in mutant animals *in vivo*. Detection of lipids *in vivo* showed that the lack of the perilipin homologue affects the intracellular distribution of lipid droplets, which is in agreement with the role of perilipin homologue LSD2 (PLIN2) in movement of lipid droplets in *Drosophila* (*Cohen, 2005*; *Welte et al., 2005*). Analysis of lipid-containing structures in developing embryos and in adult tissues suggested that W01A8.1 protein forms are likely to act differently in embryos than in adult tissues and lipid-containing structures in embryos are likely to differ from those of adult tissues. The characteristic aggregation of lipid-containing structures around the embryonic nuclei clearly detected in *C. elegans* embryos by CARS microscopy are reminiscent of lipid droplets recently reported to be a characteristic feature of cancer stem cells in colorectal carcinomas (*Tirinato et al., 2015*).

Our results suggest that the previously accepted view of a perilipin-independent nematode fatty acid flux of LDs needs to be revisited. Clearly we see evidence for perilipin-like LD regulation that is evolutionarily conserved. Based on the results reported here, *W01A8.1* is renamed with permission from WormBase as *plin-1* (*Cel-plin-1* with species identifier). With only a single gene and a toolbox of forward and reverse genetic approaches at hand, *C. elegans* offers an opportunity to explore the exact role of perlipin-related factors in fat regulation throughout development of many different somatic and germline cells. Exploitation of these opportunities will likely reveal new levels of regulation and novel players in the complex and vital regulation of fat in all organisms.

## ACKNOWLEDGEMENTS

The authors thank WormBase and NCBI for accessibility of data and bioinformatics support and CGC for the N2 wild type strain. Authors thank Dr. Zhongxiang Jiang and Leica Microsystems CMS GmbH (Mannheim, Germany) for CARS microscopy. Authors thank Dr. Sebastian Honnen, and Reviewer 2 for valuable suggestions and corrections.

### Funding

This work was supported by the European Regional Development Fund "BIOCEV—Biotechnology and Biomedicine Centre of the Academy of Sciences and Charles University in Vestec" (CZ.1.05/1.1.00/02.0109); the grant PRVOUK-P27/LF1/1 from Charles University in Prague; the grants SVV 260023/2014 and SVV 260149/2015 from Charles University in Prague. MWK is supported by the Intramural Research Program of the National Institute of Diabetes and Digestive and Kidney Diseases (NIDDK) of the National Institutes of Health, USA. The authors received funds from MediCentrum Praha a.s. to support the work reported in this publication. ZK and MK contributed personal funds to this work. The funders (excluding the authors) had no role in study design, data collection and analysis, decision to publish, or preparation of the manuscript.

### Grant Disclosures

The following grant information was disclosed by the authors:
European Regional Development Fund: CZ.1.05/1.1.00/02.0109.
Charles University in Prague: PRVOUK-P27/LF1/1, SVV 260023/2014, SVV 260149/2015.
National Institute of Diabetes and Digestive and Kidney Diseases (NIDDK).

### Competing Interests

Marta Kostrouchová is an Academic Editor for PeerJ. The authors declare there are no competing interests.

### Author Contributions

- Ahmed Ali Chughtai, Filip Kaššák and Markéta Kostrouchová conceived and designed the experiments, performed the experiments, analyzed the data, wrote the paper, prepared figures and/or tables, reviewed drafts of the paper.

- Jan Philipp Novotný conceived and designed the experiments, performed the experiments, analyzed the data, wrote the paper, reviewed drafts of the paper.
- Michael W. Krause conceived and designed the experiments, analyzed the data, wrote the paper, reviewed drafts of the paper.
- Vladimír Saudek conceived and designed the experiments, analyzed the data, wrote the paper, prepared figures and/or tables, reviewed drafts of the paper.
- Zdenek Kostrouch and Marta Kostrouchová conceived and designed the experiments, performed the experiments, analyzed the data, contributed reagents/materials/analysis tools, wrote the paper, prepared figures and/or tables, reviewed drafts of the paper.

## Human Ethics

The following information was supplied relating to ethical approvals (i.e., approving body and any reference numbers):

Human PLIN2 and PLIN3 were cloned from a collection of anonymous unmarked samples (PLIN2), and from human peripheral lymphocytes (PLIN3) donated by a volunteer with a written consent in compliance with the legislation of the Czech Republic and European Union (Act No 372/2011 of 11. 11. 2011 on Health Care Services, Coll., Paragraph 81, section 1a and section 4a, which is in accordance with the declaration of Helsinki) and was approved by the Ethical Committee of the First Faculty of Medicine, Charles University in Prague (Ref. No. MZ13-UK1LF-KostrouchZdenek).

## Supplemental Information

Supplemental information for this article can be found online at http://dx.doi.org/10.7717/peerj.1213#supplemental-information.

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
