# Peer review of "Perilipin-related protein regulates lipid metabolism in C. elegans"

_PeerJ, doi:10.7717/peerj.1213_

## Round 0.1 · original submission · Major Revisions

Dear Dr Zdenek Kostrouch,

As you will see, a number of important points have been raised by the referees, so first we would like to invite you to revise your manuscript in response to these comments before we reach a final decision on publication.

In particular, both reviewers felt that more functional experimental data, rather than mere speculation, should be provided before you can conclude that the identified peripilin-like C. elegans gene indeed acts as a true mammalian hortolog. I also agree that more detailed information on specific methodology (e.g. extensive blast search, CRISP/Cas9).

In conclusion, we feel that you will need to provide a full response to all of the comments raised, with additional data as requested, in order to make a case for further consideration. Please motivate in the rebuttal point-by-point reply letter if and why you decide not to address any specific reviewer’s concern.

·

Basic reporting

Major Comments
Line 75 “…inhibition or elimination…” – Inhibition, speaking for pharmacological treatment or similar, is not shown, only elimination by forward or reverse genetics. "Inhibition" should be deleted from the text.


Minor Comments
The authors use abbreviation, which should be explained (e.g. PAT).

Line 52 “…access to various proteins to stored fat…” – in which way? Please clarify.

Line 66 The authors should show at least a domain model of C. elegans W01A8.1 in comparison to human and drosophila proteins! Better also a 3D reconstruction. Using the simple web based software swissmodell.expasy.org it is possible to produce 3D reconstructions of W01A8.1 and human Perilipin which show some obvious similarity.

Line 87 “searches” has to be “Searches”

Lines 166 – 172 This protocol is not used and should be omitted in the manuscript.

Lines 192 – 193 Nile red was not used here. All figures that are shown where produced using LipidTox.

Lines 243 -244 Here the authors should add a citation underlining the association of perilipins and lipid droplets as well as providing immunhistochemical images which are comparable to the GFP images produced here, e.g. “..as shown before (Kozuko et al., 2015).) Diabetes. 2015 Jan;64(1):299-310. doi: 10.2337/db14-0104. Epub 2014 Aug 11. “Clinical and molecular characterization of a novel PLIN1 frameshift mutation identified in patients with familial partial lipodystrophy.” Kozusko K1, Tsang VH2, Bottomley W3, Cho YH4, Gandotra S5, Mimmack M1, Lim K1, Isaac I1, Patel S1, Saudek V1, O'Rahilly S1, Srinivasan S6, Greenfield JR7, Barroso I3, Campbell LV8, Savage DB9.

Line 283 “…in vitro…” should be generally italic as well as “…in vivo…”

Line 286 “… gonad anomalities…” should be “gonadal abnormalities”

Line 298 The term “N2” should only be used in the methods section where it is defined to be the used wild type. Throughout the manuscript “N2” should be replaced by “wild type”.

Figure 5: Mistake in heading “(Lass et al., 20114)” as well as “…C.elegans…” where a space is missing.

Experimental design

Major comments
Line 95 The co-injection marker is not shown anywhere! The authors should provide more supplementary data of controls as well as the produced strains should be send to the CGC to be available for other researchers.

Lines 156 – 160 The protocols are non-standart and in-conclusive. Why using two different protocols? To determine the whole progeny/individual and plot in a scatter plot would have been more informative. I recommend to perform a standart fertility assay determining the number of progeny of individual worms (10/group) as well as lethality within these groups at the same time.
There is no quantification for embryonic lethality so far, which should be added.

Minor comments
Lines 152 – 154 Standart protocol would be OD600=0.9 and a defined volume per plate and not just “…cover the surface and (remove) excess…”. The density and number of bacteria is critical for RNAi effect. It would have been better to use a fixed volume for all plates of bacteria OD600=0.9

Lines 156 – 160 Which Temperatur was used?

Line 259 If technically not to complicated it would have been a good idea to produce strains expressing both, human PLIN1::GFP (PLIN2::GFP) and maybe W01A8.1::mCherry to make colocalization studies. This could be interesting in the future for following studies (also regarding 267 – 268).

Validity of the findings

Major comments
Lines 97 – 103 They do not show a proof for the succesul mutagenization by CRISP/Cas9 (only the strategy). Single Worm PCR methods are described, but not used for the proof – why? This validation should be in the supplementary material


Line 255 The authors state that animals expressing W01A8.1 a/c::GFP populations have generally low fat content, but this is not measured/quantified! How do they know? This should be clarified and images/quantification should be presented at least as supplementary figures.

Lines 272 – 275 The used protocol to determine progeny is not apparent. What stand “n=260/550/1000/1400” for? Did the authors determine the progeny for 1400 independent adult hermaphrodites? From which numbers is “…1/3 less progeny…” calculated? The authors state a “…a significantly smaller brood size…” but no statistical test was performed or it is not well annotated.
This passage needs to be reworked carefully! Also consider the suggestion to perform standard fertility assay (confer e.g.: “Friedman D, Johnson T (1988) A mutation in the age-1 gene in Caenorhabditis elegans lengthens life and reduces hermaphrodite fertility” or Honnen et al., 2012 Plos One.)

Line 286 The authors state that there are gonadal abnormalities and low brood size in strains with high levels of W01A8.1(a)synth::GFP. This raises again some questions. Are there more than one such strain or different lines? Where they sorted in a way to determine lines with higher levels to determine gonadal abnormalities and brood size within these populations? If yes, how and where are the methods annotated as well as images and quantification of gonadal abnormalities moreover statistics to compare brood size between different lines/strains?
It has to be clarified if this is only an observation!

Lines 290 – 292 Again, the authors speculate about consequences of W01A8.1 locus abundance using data that is not in the manuscript!

Lines 285 – 294 This passage also needs to be reworked carefully stating which statements refer to data and which arise from observation. What are results and what is speculation? In general the passage could be very informative and interesting, but in the current status it seems to be misleading and technically not valid. The data which is discussed and/or used to produce graphs should be shown in tables in the supplementary section.


Minor comments
Lines 76 – 78 inconclusive with regard to very early embryonic stages! Which additional is it? Why is there no GFP in very early embryos?

Lines 250 – 251 Why are there phenotypes through loss of W01A8.1 before the three fold stage, but the GFP signal get visible not until three fold stage? This should be discussed.

Lines 277 . 278 As Perilipins should protect LD from Lipolysis it is counterintuitive in the first place that there are “…larger lipid droplets…”. The Authors should correlate W01A8.1 activity with overall fat content (which should be reduced) and lipid droplet size (which seems to be enlarged) by quantifying both parameters carefully. This should also be discussed with regard to the so far not cited literature “PERILIPIN-Dependent Control of Lipid Droplet Structure and Fat Storage in Drosophila ” by Beller et al., 2010 (doi:10.1016/j.cmet.2010.10.001), as they also observed larger LD in Drosophila Perilipin knock out animals (more specific PERILIPIN1 in contrast to the Drosophila double mutants [PERILIPIN1 and PERILIPIN2 knock outs] used by Bi et al., 2012).

Line 306 The authors speak about “…larger LD…”, but the quantification is missing. It should be easy to use existing images to quantify LD size under different conditions and show a scatter plot at least in the supplementary section.

Lines 306 – 308 It is confusing that the null mutant strain shows only a 10% decrease in total number of progeny whereas the knock down population shows 1/3 decrease in progeny. This should also be discussed and commented.

Additional comments

Lines 327 – 328 In future studies co-localization analysis of these factors would be interesting.

Lines 363 – 369 This is a very good point and I absolutly agree!

If the W01A8.1 lines are stil available the authors should follow the abundance (whole fluorescence using e.g. ImageJ) while starving the animals or compare starved and non-starved populations at a given time point. This would be a functional read out with regard to W01A8.1 in association with LDs. The expression of Perilipins during starvation is discussed in the literature and it would be of interest how the proposed homologue behaves with regard to nutrition status.

The model is not easy to comprehend, because it consists only of abbreviations. If the authors want to draw this model they should try to make it more understandable through reducing the number of abbreviations.

In general it is very interesting work and the body of results, especially genetic work, is convincing, but in my opinion the authors should work at least on the major comments with regard to the observations and functional read outs (quantification and statistical tests) for the final publication.

Reviewer 2 ·

Basic reporting

No comments

Experimental design

No comments

Validity of the findings

Kostrouch et al., describe in their present manuscript the identification and initial characterization of a candidate PERILIPIN homolog in the nematode C.elegans. So far, C. elegans was thought to lack any PERILIPIN like protein(s). The authors, however, used elaborate BLAST methodology to identify also more hidden homologies and uncovered the gene W01A8.1 as a potential PERILIPIN relative. Subsequently, the authors expressed the identified protein tagged with GFP in worms and compared its subcellular localization to human PERILIPINS 1 and 2 expressed in the same way. In the following, the authors present data describing the phenotypes observed when W01A8.1 is knocked-down by RNAi or the phenotype of worms which are deficient for the W01A8.1 encoding gene due to a deletion. Such worms show an altered lipid droplet morphology in embryos and a slightly reduced fecundity.

The paper is to the most part well written and touches a highly discussed topic. Conclusions, however, appear to go beyond what the data supports. Thus, the following points should to be addressed to allow publication:

- The similarity/homology of W01A8.1 to PERILIPINs is weak. In order to make the claim that the identified protein is a true (functional) PERILIPIN homolog further experiments are needed. Due to the non-obvious sequence similarity the details of the bioinformatics methodology used by the authors should also be explained in a way that a layperson can also understand what was done and how this diverges / is similar to a regular BLAST search. A true functional similarity could be shown, for example, by demonstrating a meaningful interaction between W01A8.1 with known PERILIPIN interactors such as the worm homologs of ATGL, CGI-58 or HSL, for example. The authors suggest a similar function of W01A8.1 and the PERILIPINs based on such interactions in their last figure and this appears to be too speculative at this point as far as I can see (as well as it is not clear whether the suggested similarity to PERILIPIN1 would make sense based on the W01A8.1 loss-of-function phenotype; see below). In my opinion, also the renaming of W01A8.1 to Cel-plin-1 (line 239) asks for additional data and is not justified at the present point of time. Thus, the authors will need to tone down their statements and explain in greater detail the similarities or - in order to stick to the current depth of functional classification – need to perform additional experiments, as for example, the ones suggested by this reviewer.
- The phenotype of the W01A8.1 loss-of-function also appears to be weak for a PERILIPIN homolog unless there is redundancy. In flies or mammals loss-of individual PERILIPIN genes show much more prominent phenotypes (as long as there is no redundancy). In flies, however, loss-of both PERILIPIN gene copies - flies encode only two PERILIPINs – also result in a somewhat intermediate lipid storage phenotype and a reduced dynamics of lipid remobilization and restoration. Is such a phenotype also present in the mutant worms? I think the authors should discuss their findings also in the light of the Drosophila PERILIPIN double loss-of-function mutant (Beller et al., Cell Metab 2010). Perhaps the protein W01A8.1 identified is the homolog of the ancestry lipid regulatory system acting independent of classical PERILIPINs proposed in the 2010 Cell Metabolism paper? In my opinion this reference is better suited as the one used by the authors (Bi et al., 2012).
- In lines 218/219 the authors state that: “…in stark contrast to the chordates proteomes that had 2 to 5 perilipin paralogues.” This statement appears imprecise, as insects, for example encode 2 PERILIPINs, whereas chordates such as mammals encode 5 PERILIPINs. Thus, the authors need to change chordates with something more appropriate.
- The localization of W01A8.1 to LDs is shown in a sufficient way. However, the authors state in lines 254 to 257 that “The animals expressing W01A8.1a/c::GFP had generally low fat content, keeping with the expected overexpression of the native isoform b from this transgene. We also noted that animals with high levels of W01A8.1a/c::GFP had an altered morphology of the gonad and embryos.” The authors need to present data supporting these notions or remove them. An altered lipid storage content would indeed be expected for a PERILIPIN homolog. Thus, the authors should present lipid storage amount quantifications of overexpressing and mutant worms.
- The similarity of localization of the mammalian PERILIPINs and the worm protein (as stated in the paragraph headline “Human PLINs 1and 2 label identical compartments as W01A8.1 proteins in C. elegans” is not completely clear. A double transgenic worm demonstrating clear-cut co-localization needs to be shown to make the claim of localization to identical compartments.
- RNAi and control conditions shown in Fig. 4 A&B / D&E look very different for a non C.elegans person (embryo in animal and free egg?). The authors should use images more comparable or state what the differences are.
- As far as I can see there are no wildtype reference images shown for Fig. 4 H&I. If panels D&E of Fig. 4 should suit this purpose, this should be noted for the sake of clarity. All of these loss-of-function phenotypes are represented by singular images. The authors need to provide any information concerning the penetrance of the observed phenotype(s) or in the best case provide a quantification of the phenotype such as the size increase of LDs as per cent of control, for example.
- In Fig. S4 the authors show a reduction of the brood size by 10% in the W01A8.1 null mutant strain. Is this finding statistically significant and biologically relevant or are changes in this range not also present among different genotypes / strains? Additionally, this reduction of the brood size is milder as compared as the one following RNAi knockdown (30% as stated in lines 273/274 and Fig. S3). Do the authors have an explanation for this difference based on the type of perturbation?

---

## Round 0.2 · Minor Revisions

Dear Dr Zdenek Kostrouch,

we believe your revised manuscript is now strongly improved. However, as you will read, the second reviewer expressed that some of the original concerns were not appropriately addressed. Therefore, before we can accept your manuscript for publication we would like to encourage you to address at least points (i), (iii) and (iv) of the second referee. This should also clarify some minor points we felt not clearly addressed from the first reviewer (i.e. point 20). I recommend that you revise your main conclusions also in light of the 2nd referee’s interpretation of your results.

We hope our decision will not affect your future resubmission of an improved manuscript to the journal and we look forward to receive a re-revised version.

·

Basic reporting

No comments

Experimental design

No comments

Validity of the findings

No comments

Additional comments

I m satisfied with the comments, changes and especially additional experiments.

Thank you for your elaborate replies.

Reviewer 2 ·

Basic reporting

No comments.

Experimental design

No comments.

Validity of the findings

The authors somewhat improved the manuscript, and targeted some of my concerns raised during the initial review. However, several issues remain. The following points need to be clarified and/or addressed:

(i) The authors state in the rebuttal letter that a co-localization / an interaction between their potential Perilipin ortholog and the C.elegans homologs of known mammalian Perilipin interactors, such as the ATGL or HSL lipases, could not be investigated due to overexpression phenotypes or other complications. What about performing RNAi epistasis experiments? What happens in different single knock-down situations as well as combinations thereof? In case the potential ortholog plays a role in lipid deposition / lipolysis regulating processes - as drawn in the schematic Fig. 7 - it should be possible to show this by such experiments.

(ii) The CARS experiments are valuable, but do not add that much information as compared to the initial manuscript data where LipidTOX stainings were (and still are) used to justify similar conclusions. Yes, the animals are in the unfixed and unstained state and most importantly alive. However, a biochemical quantification of lipid storage phenotypes by thin layer chromatography or enzymatic assays would have been much more advantageous in the line of statistical soundness of the findings - at least in terms of the quantitative lipid storage phenotype. For the cellular / morphological phenotype, CARS is indeed the method of choice and a very welcome improvement of the study. However, the description of the analysis also shows shortcomings as described in the following point.

(iii) The new figure 6 shows a quantification of the CARS experiment results. Morphometric data are additionally represented in the form of two supplemental tables. Unfortunately, neither the legend of Fig. 6 nor the main text links to the supplemental tables, which themselves lack any explaining information e.g. in the form of a readme file, which should be added. Several statements made in the figure legend additionally need to be further explained and/or corrected: 1) " ...recognized by ImageJ program with settings yielding the biggest number of individually recognizable structures." Which settings were used? The methods section also does not provide additional information. This needs to be clarified/added. 2) "C) The total area of all CARS positive structures in single focal plane images acquired from seven representative mutant and wild type embryos." I guess the number of animals investigated holds also true for panels A & B? This is not mentioned and a reader should not need to guess what the figure actually shows. In the corresponding table, mutant animal embryos are labeld with 1a, 1b, 2, 3, 4a (why the data from this embryo was not used in the analysis lacks an explanation), 4b, 5a and 5b. A similar numbering is used for the other genotypes. What do these numbers mean? Are these indeed different animals? With the description of the data as is, a reader has a hard time to interpret the data. Given the large standard deviations the measurements appear to be quite diverse. While the shown examples might be statistically significantly different, more measurements would strengthen the author's point(s) or the way the measurements are done need to be re-evaluated. (3) The authors further write that "D) The total number of individually recognizable CARS positive structures (labeled lipid containing structures, LCS) in the two analyzed embryos...." What does this mean? Are really only two animals analyzed? For the whole part, the data description needs to be done much more careful and in a much more comprehensive manner.

(iv) The authors speculate that the short term perturbation causes more pronounced effects on the fecundity as compared to the true loss-of-function, where compensatory mechanisms could dampen the phenotype. For me, this is speculation. What about e.g. the possibility that the RNAi has detrimental off-target effects? As far as I can see, there is additionally no quantification of the RNAi efficacy shown. This should be included for the sake of proper scientific standards. The authors refer to Fig. S4 to support their interpretation, which I can not understand the way it is presented since there is no information provided with the supplemental figure. At least the ordinate description / units need to be added as is true for Fig. S5.

Additional comments

While the additional data and explanations concerning e.g. the bioinformatics part or the CARS experiments, add some further support for the author's notion that the newly identified protein might acts as a Perilipin ortholog, there are still shortcomings of the manuscript as mentioned above and for me it is still not unambigous whether the protein really needs to be reclassified / should be reclassified as "the" C.elegans Perilipin or whether such a reclassification is justified.
Instead, I would like to suggest a different point of view concerning the ancient role of Perilipin-like/Perilipin-related proteins. The authors might want to consider an interpretation of their findings also in this direction: First, Perilipin double mutant flies do not show small lipid droplets, as mentioned by the authors in line 424, but rather show the unilocular lipid droplet phenotype of perilipin 1 deficient flies. Further, perilipin double mutant flies develop more or less wildtype lipid storage levels upon maturation. Since the lipid remobilization and restoration of lipid stores is slowed down in the perilipin double mutant flies as compared to control animals, another regulators system might exist, which appears to act independent of the Perilipins. I think the protein identified by the authors could represent such an ancient Perilipin-like protein and instead of stressing a potential hidden similarity to the Perilipins, the authors could also argue in such a direction. This would explain why the sequence similarity to Perilipins is not obvious, while the protein could be involved in Perilipin-like functions.

---

## Round 0.3 · accepted · Accept

Dear authors,

The re-revised manuscript has been substantially improved in light of the latest reviewers comments and the manuscript is now suitable for publication.